# Expression, Prognostic Value and Correlation with HPV Status of Hypoxia-Induced Markers in Sinonasal Squamous Cell Carcinoma

**DOI:** 10.3390/jpm13050767

**Published:** 2023-04-29

**Authors:** Alessandro Vinciguerra, Vincent Bedarida, Charlotte Pronier, Sophie El Zein, Michel Wassef, Sarah Atallah, Florian Chatelet, Joffrey Molher, Philippe Manivet, Philippe Herman, Homa Adle-Biassette, Benjamin Verillaud

**Affiliations:** 1Otorhinolaryngology and Skull Base Center, AP-HP, Hôpital Lariboisière, 75010 Paris, France; 2Centre de Ressources Biologiques Biobank Lariboisière (BB-0033-00064), DMU BioGem, AP-HP, 75010 Paris, France; 3Université Rennes, CHU Rennes, Virology, Inserm, EHESP, Irset (Institut de recherche en santé, environnement et travail) UMR_S 1085, F-35000 Rennes, France; 4Pathology Department, Institut Curie, 75010 Paris, France; 5Pathology Department, DMU DREAM, AP-HP, Hôpital Lariboisière, 75010 Paris, France; 6Université Paris Cité, 75010 Paris, France; 7INSERM UMR 1153 ECSTRRA Team, 75010 Paris, France; 8INSERM U1141 NeuroDiderot, 75010 Paris, France

**Keywords:** squamous cell carcinoma, immunohistochemistry, overall survival, hypoxia, hypoxic markers

## Abstract

(1) Background: In head and neck squamous cell carcinoma, tumor hypoxia has been associated with radio/chemoresistance and poor prognosis, whereas human papillomavirus (HPV)-positive status has a positive impact on treatment response and survival outcomes. The aim of this study was to evaluate the expression and the potential prognostic value of hypoxia-induced endogenous markers in patients treated for squamous cell carcinoma of the nasal cavity and paranasal sinuses (SNSCC), and their correlation with HPV status. (2) Methods: In this monocentric study, patients treated in a curative intent for a SNSCC were screened retrospectively. Protein expression of CA-IX, GLUT-1, VEGF, VEGF-R1, and HIF-1α was determined by immunohistochemical staining, scored, and then correlated with overall survival (OS) and locoregional recurrence free survival (LRRFS). HPV status was assessed and correlated with hypoxic markers. (3) Results: 40 patients were included. A strong expression of CA-IX, GLUT-1, VEGF, and VEGF-R1 was detected in 30%, 32.5%, 50%, and 37.5% of cases, respectively. HIF-1α was detected in 27.5% of cases. High CA-IX expression was associated in univariate analysis with poor OS (*p* = 0.035), but there was no significant association between GLUT-1, VEGF, VEGF-R1, and HIF-1α expression, and OS/LRRFS. There was no correlation found between HPV status and hypoxia-induced endogenous markers (all *p* > 0.05). (4) Conclusions: This study provides data on the expression of hypoxia-induced endogenous markers in patients treated for SNSCC and underlines the potential role of CA-IX as a prognostic biomarker for SNSCC.

## 1. Introduction

Sinonasal squamous cell carcinoma (SNSCC) is the most common sinonasal malignancy, making up approximately 60% of cases [1]; an incidence rate of 0.36 cases per 100,000 population per year has been reported [2]. Because of its low chemo/radiosensitivity, the therapeutic strategy is generally based on surgery combined with radiotherapy [3]. Nevertheless, despite improvements in endoscopic endonasal surgery and modulated radiotherapy, in the last thirty years, the overall survival rate at five years has remained stable at around fifty-three percent [3,4]. In this context, it seems important to investigate the potential biological mechanisms that may be associated with a resistance to chemo/radiotherapy. Tumor hypoxia has been shown to have a negative impact on chemo/radiosensitivity and on survival in several tumor types, including HNSCC [5]. In fact, the lack of oxygen induces a reduction of reactive oxygen species, significantly decreasing the DNA damage caused by radiotherapy, and increases the proliferation of anarchist blood microvessels, causing irregular drug diffusion [6]. Although significant progress has been made in the understanding of the biology underlying sinonasal cancers [7], very few studies have been published on the hypoxic changes in SNSCC [8].

Several factors are involved in the cellular response to hypoxia and can be assessed using standard immunohistochemistry procedures. Hypoxia-inducible factor 1 (HIF-1) plays a major role in the regulation of hypoxic response element transduction [9]. HIF-1 is a heterodimer composed of HIF-1α, the expression of which is induced by hypoxia, and HIF-1β which is constitutively expressed in the nucleus independently of hypoxia. Under normoxic conditions, after hydroxylation, HIF-1α binds with the von Hippel-Lindau protein to cause its degradation. In hypoxic conditions, HIF-1α translocates to the nucleus and binds to HIF-1β to induce the expression of over 40 hypoxia-related genes [10]. These include: transmembrane carbonic anhydrase IX (CA-IX), which codes for a protein implicated in the adaption of the hypoxic microenvironment, glucose transporter 1 (GLUT-1) which allows glucose transport, and vascular endothelial growth factor (VEGF) and vascular endothelial growth factor-receptor 1 (VEGF-R1), implicated in neo-angiogenesis. 

On the other hand, human papillomavirus (HPV) positive status is associated with a better response to radiotherapy, and with better survival outcomes in HNSCC, especially in the oropharynx [11,12]. The implication of HPV in the pathogenesis of SNSCC has already been reported [13]. A recent study based on the surveillance, epidemiology, and end Results Program (SEER) database even found an increase in the incidence of HPV-associated SNSCC and the prevalence of HPV-positive SNSCC over recent decades [14]. The prognostic significance of HPV-status in SNSCC has remained uncertain for a long time, but it now seems that HPV positivity is associated with a better overall survival [11,15]. The potential association between hypoxia and HPV status has, however, never been explored.

The aim of this study was to explore the expression and the prognostic impact of five markers of hypoxia: HIF-1α, CA-IX, GLUT-1, VEGF, and VEGF-R1 in a group of patients treated for sinonasal squamous cell carcinoma, and to analyze their correlation with HPV status.

## 2. Materials and Methods

All the patients treated in a curative intent for a SNSCC at a tertiary care center between January 2009 and December 2020 with a minimum follow-up of 18 months were included in the study. Patients with missing clinical data or with no pathological specimens available were excluded.

Data collection was approved by the French authorities (CNIL No. 2226104); informed consent was obtained from all subjects. Patient records and information were anonymized and de-identified prior to analysis. The following clinical data were retrospectively collected from the hospital files and records: age, gender, smoking status, TNM staging according to the 8th UICC, treatment strategy, and survival outcomes.

### 2.1. Treatment Modality

The treatment strategy was decided by the oncological hospital board according to the national recommendations (REFCOR). Briefly, all patients with a removable tumor underwent trans-facial open surgery or endoscopic surgery, followed by radiotherapy. Adjuvant chemo-radiotherapy was carried out in cases of surgical margin less than 5 mm; tumor invasion of the orbital apex, cavernous sinus, and of the brain was considered as contraindication for surgery and treated with chemoradiotherapy, similar to patients who refused surgery. Neoadjuvant chemotherapy was considered in locally advanced tumors that were not amenable to surgery or radiotherapy at the outset, with the aim of achieving tumor deflation and facilitating surgery or radiotherapy/chemoradiotherapy. In cases of partial/complete response to neoadjuvant chemotherapy, chemoradiotherapy was considered; in cases of stable or progressive disease, surgery was considered, or if this was impossible, chemoradiotherapy.

### 2.2. Immunohistochemistry

Formalin-fixed, paraffin-embedded (FFPE) tissue specimens were obtained and handled by standard histological procedures. The pathological analysis was performed on the surgical specimen in operated patients, and on the biopsies in patients treated by chemo-radiotherapy. Serial 4-μm sections were prepared. For CA-IX, GLUT-1, VEGF, and VEGF-R1, tissue sections were microwaved at 98 °C for 30 min in citrate buffer (10 mM, pH 7.3), and then incubated with the primary antibody (anti-human CA-IX NB100-417, Novus Biologicals (Centennial, CO, USA); anti-human GLUT-1 cbl242, Merck Millipore; anti-human VEGF abs82, Merck Millipore; anti-human VEGF-R1 ab2350, Abcam, respectively). Binding of the primary antibody was detected with the CSA II kit from DAKO (based on a tyramide amplification system; DAKO, Carpinteria, CA, USA). For HIF-1α, slides were treated with target retrieval solution (DAKO) at 97 °C for 45 min and then incubated with primary antibody (antihuman HIF-1α 61095, BD Transduction Laboratories). Binding of the primary antibody was detected with the Catalyzed Signal Amplification System (DAKO). For all cases nuclei were counterstained with hematoxylin. 

The immunohistochemical expression of hypoxic markers was assessed by 3 independent readers. When independent assessments of a case differed, the case was rechecked, and the final score was determined by consensus. The expression of CA-IX, GLUT-1,VEGF, and VEGF-R1 was classified as follows: −, no staining; 1+, nuclear staining in less than 1% of cells; 2+, nuclear staining in 1–10% of cells and/or with weak cytoplasmic staining; 3+, nuclear staining in 10–50% of cells and/or with distinct cytoplasmic staining; and 4+, nuclear staining in more than 50% of cells and/or with strong cytoplasmic staining. HIF-1α staining was considered positive in cases of nuclear detection of the protein. 

### 2.3. HPV Status 

HPV genotyping was used to assess HPV status. Formalin-fixed, paraffin-embedded (FFPE) biopsies were deparaffinized, and HPV detection and genotyping was carried out by multiplex real-time PCR using the Anyplex II HPV28 Detection assay (Seegene, Seoul, Republic of Korea). This assay allows simultaneous detection and identification of 28 HPV types including 13 high-risk. 

### 2.4. Statistical Analysis

The Chi-square test or Fisher’s exact test for independent proportions was performed to evaluate the differences and similarities in expression of CA-IX, GLUT-1, VEGF, VEGF-R1, HIF-1α, and in HPV status. Overall Survival (OS) and locoregional recurrence free survival (LRRFS, time to locoregional disease progression following complete tumor regression) curves were plotted using the Kaplan–Meier method. Statistical significance was set at *p* < 0.05.

## 3. Results

A total of 40 patients were included in the study. The mean and median follow-up were 28.4 months (SD 19.8, 18–77 months), and 27 months, respectively. The clinical characteristics are summarized in Table 1; 57.5% of the patients were >60 years-old, and 70% were male; 52.5% of cases were smokers. The nasal fossa was the most common area of tumor (27.5%). Most patients presented with advanced stage tumors: 67.5% of patients had a T4-stage tumor. There were no distant metastases, but 7.5% of patients presented with nodal involvement at diagnosis. Among the treatment modalities (Table 2), the most frequent was surgery followed by radiotherapy (37%).

### 3.1. Immunohistochemical Detection of Hypoxic Markers

The results of hypoxic marker detection is summarized in Table 3. A strong expression (4+) of CA-IX and GLUT-1 was detected in 30%, and 32.5% of cases, respectively, with a higher level in malignant cells located peripherally to the necrotic zones. Illustrative cases are displayed in Figure 1. Tumors with a high CA-IX expression correlated with poor OS (*p* = 0.035) (Figure 2). There was no correlation found between strong GLUT-1 expression and OS (*p* = 0.51; Appendix A).

VEGF and VEGF-R1 were detected at high levels in 50% and 37.5% of cases, respectively; there was a tendency to a decreased OS in VEGF-positive tumors (*p* = 0.14), but no correlation was found between VEGF-R1 expression and OS (*p* = 0.79) (Appendix A). HIF-1α was detected in 27.5% of cases, and there was no correlation determined with OS (*p* = 0.59, Appendix A).

None of the hypoxic markers were found to have correlated with LRRFS. The Spearman’s coefficient between the markers was > 0.05, underlying the absence of direct associations between HIF-1α, CA-IX, GLUT-1, VEGF, and VEGF-R1.

### 3.2. Correlation between HPV Status and Hypoxic Markers

The HPV status was positive in 27.5% (11/40) of cases, including high risk HPV-16 (3 cases, 27.5%), and HPV-18 (1 case, 9%), and low risk HPV-82 (4 cases, 36.5%), and HPV-45, −68, −6 (1 case, 9%). There was no statistical correlation found between HPV status and OS/LRRFS (*p* = 0.62 and 0.43, respectively; Appendix A).

In addition, there were no statistical associations found between the HPV status and CA-IX, VEGF, VEGF-R1, GLUT-1, and HIF-1α (all *p* > 0.05).

## 4. Discussion

In this retrospective study of forty patients treated for a SNSCC, a strong expression of the hypoxic markers CA-IX, GLUT-1, and VEGF-R1 was detected by immunohistochemistry in approximately a third of the patients. High levels of VEGF were detected in 50% of the cases, while HIF-1α was detected in 27.5% of the cases. CA-IX overexpression was associated with a poor OS in univariate analysis, but there were no significant associations found between the other markers and OS/LRRFS. Moreover, there was no association found between the HPV status and the expression of hypoxic markers. To the best of our knowledge, there has been no previous study focusing on the expression of hypoxic markers and their correlation with HPV in SNSCC.

Hypoxia is a frequent feature of head and neck squamous cell carcinoma, and has been associated with radio/chemoresistance, a higher metastatic potential, and a poor prognosis [16]. In this study, hypoxia was investigated by immunohistochemical assessments of its main markers, namely CA-IX, GLUT-1, VEGF, VEGF-R1, and HIF-1α. However, over the last decades, other methods have been used to measure hypoxia, such as polarographic pO2 histography, or positron emission tomography (PET) imaging with hypoxia-specific radiotracers ([18F]fluoromisonidazole ([18F]FMISO)) [17,18]. The main advantages of immunohistochemistry are the simplicity, the reliability, and the relatively low cost of the technique, as well as the potential of marker co-detection, and the possibility of repeatedly detecting and monitoring the biomarkers.

HIF-1α plays a pivotal role in the cellular response to hypoxia and has been demonstrated to be an independent negative prognostic factor in head and neck SCC [19,20]. These results are, however, in contrast with our observations, which showed no significant association between prognosis and HIF-1α expression; the absence of this association can be explained, as already stated, by the fact that HIF-1α is generally only transiently induced, and then undergoes rapid degradation and partial feedback inhibition. This protein is therefore difficult to detect by immunohistochemistry [21], and its precise prognostic signification is still to be determined.

CA-IX and GLUT-1 represent the main HIF-1α induced proteins and cooperate to allow the survival of cancer cells under hypoxic conditions. Specifically, the rapid proliferation of malignant tumors requires high levels of energy, which are generally obtained from glycolysis. GLUT-1 plays a role in this metabolic reprogramming, allowing an increased glucose uptake and a higher rate of anaerobic glycolysis which, on the other hand, also increases the intracellular pH [22]. Nevertheless, the expression of CA-IX maintains the intracellular pH homeostasis by protecting the cells from intracellular acidification, thereby producing an extracellular acidic setting [23]. This tumor acid environment causes several clinical effects such as: decreasing drug uptake in tumor cells due to the altered protonation state, selecting highly aggressive and drug-resistant phenotypes with stem cell characteristics, enhancing invasion and metastatic processes, inducing VEGF angiogenesis, and decreasing immune infiltration of the tumors, leading to the development of a more malignant phenotype [24]. CA-IX has been described as a promising endogenous marker of hypoxia [25,26], and a recent meta-analysis confirmed its prognostic role on overall and disease-free survival in head and neck malignancies, without, however, any sub-analysis of the different localizations, tumor size, grade, and nodal status [27]. The significance of GLUT-1 overexpression is more debated; some authors report the absence of association with survival [16], while others suggest that it has a prognostic value [22]. The prognostic significance of CA-IX was confirmed in our study which specifically focused on SNSCC, but this was not the case for GLUT-1. 

VEGF is another hypoxia-responsive factor that is a key player in the development of tumor growth as it promotes angiogenesis, which is a prerequisite for tumor aggressiveness [28] and has been linked to poor prognosis, nodal metastasis, and low survival in head and neck squamous cell carcinoma [25,29]. This is in contrast with the results of the current study. This may be explained by the fact that the relatively small size of the cohort limits the power of the study. Of note, although a meta-analysis performed by Zang et al. [28] found that VEGF overexpression was significantly associated with a worsened overall and disease-free survival in head and neck squamous cell carcinoma, the authors underlined that these results were obtained only using univariate analysis. Previous studies have also failed to confirm the association between VEGF expression and survival outcomes [30].

In oropharyngeal squamous cell carcinoma, it has been well established that HPV-positive tumors were more radiosensitive and had a better prognosis compared to HPV-negative tumors [31]. This is in contrary to hypoxic tumors, which are less radiosensitive and are associated with a very poor prognosis [16]. The link between HPV status and hypoxia remains, however, unclear [12,14]. Clinical data suggests that there is no significant difference in the level, nor distribution of hypoxia in HPV-positive and HPV-negative head and neck squamous cell carcinoma, as measured by a 15-gene hypoxia classifier and 18F-MISO PET [32,33]. This is in line with our results, which suggest that there is no correlation between the HPV status and hypoxic markers in SNSCC. Of note, HPV status was assessed by genotyping; although p16 is a surrogate of HPV status in oropharyngeal squamous cell carcinoma [19], it has been shown that it is not a performant tool to assess HPV status in SNSCC [11]. Another option would have been to assess the expression of the oncogenic E6 and E7 RNA encoded by HPV, either using in situ hybridization or RT-PCR, with the advantage of these techniques is being able to confirm not only the presence of HPV in tumor cells, but also to its transcription [34,35].

This study has several limitations: first, the retrospective nature of the data; second, the relatively limited size of the cohort, and the heterogeneity of the treatment strategies that did not permit a multivariate analysis; and third, the short follow-up (28.4 months), that may have had an impact on the survival analysis. 

## 5. Conclusions

In this retrospective study, the expression of several hypoxic markers was assessed by immunohistochemistry in a series of 40 patients treated for a SNSCC. A strong expression of the hypoxic markers CA-IX, GLUT-1, VEGF, and VEGF-R1 was found in 30%, 32.5%, 87.5%, and 37.5% of cases respectively. HIF-1α was detected in 27.5% of cases. High CA-IX expression was associated with poor OS (*p* = 0.035) in univariate analysis, but there were no significant associations found between GLUT-1, VEGF, VEGF-R1, and HIF-1α expression and OS/LRRFS. Finally, there was no correlation found between the HPV status and the hypoxia-induced endogenous markers (all *p* > 0.05).

## Figures and Tables

**Figure 1 jpm-13-00767-f001:**
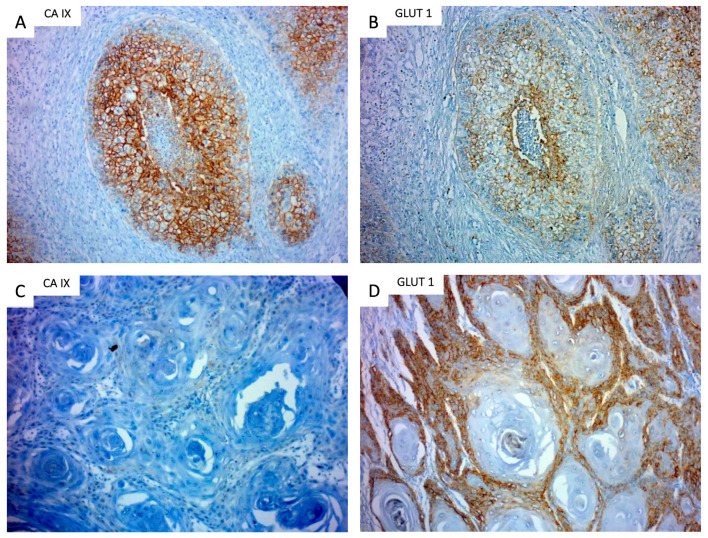
Examples of CA-IX and GLUT-1 expression. (**A**) High expression (4+) of CA-IX; (**B**) moderate expression (3+) of GLUT-1 surrounding the necrotic area; (**C**) very low expression of CA-IX (1+); and (**D**) intense expression of GLUT-1 (4+).

**Figure 2 jpm-13-00767-f002:**
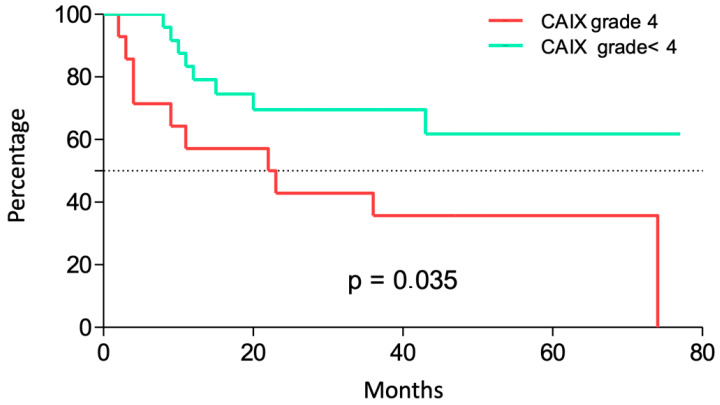
Kaplan–Meier survival curve showing overall survival based on CA-IX expression.

**Table 1 jpm-13-00767-t001:** Clinical characteristic of the enrolled patients. TNM staging according to the 8th UICC.

		N (%)
Age (mean)	<60 years	17 (42.5)
	>60 years	23 (57.5)
Sex	Female	12 (30)
	Men	28 (70)
Smoker	Yes	21 (52.5)
	No	19 (47.5)
Localization	Nasal fossa	22 (54)
	Ethmoid sinus	9 (23)
	Maxillary sinus	9 (23)
T staging	T1	2 (5)
	T2	6 (15)
	T3	5 (12.5)
	T4	27 (67.5)
N staging	0	37 (92.5)
	1	1 (2.5)
	2a	1 (2.5)
	2b	1 (2.5)
M staging	0	40 (100)
	1	0

**Table 2 jpm-13-00767-t002:** Treatment modalities and results at the last follow-up. CT, chemotherapy; RT, radiotherapy; AWD, alive with disease; DOD, dead of disease.

	N (%)	AWD, N (%)	DOD, N (%)
Surgery alone	6 (15)	1 (16)	2 (33)
Surgery + RT	16 (40)	1 (6)	4 (26)
Surgery + adjuvant RT-CT	4 (10)	0	2 (50)
Induction CT + surgery + adjuvant RT	6 (15)	0	3 (50)
Induction CT + RT	4 (10)	1 (25)	2 (50)
RT + CT	4 (10)	0	4 (100)

**Table 3 jpm-13-00767-t003:** Expression of hypoxic markers in the enrolled patients.

	CA-IX	VEGF	VEGF-R1	GLUT-1	HIF-1α
Positive	34 (85%)	34 (85%)	30 (75%)	25 (62.5%)	11 (27.5%)
1+	4 (10%)	1 (2.5%)	0	2 (5%)	-
2+	8 (20%)	7 (17.5%)	12 (30%)	6 (15%)	-
3+	10 (25%)	6 (15%)	3 (7.5%)	4 (10%)	-
4+	12 (30%)	20 (50%)	15 (37.5%)	13 (32.5%)	-

## Data Availability

The data presented in this study are available on reasonable request from the corresponding author.

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
