# Peer review of "Expression, Prognostic Value and Correlation with HPV Status of Hypoxia-Induced Markers in Sinonasal Squamous Cell Carcinoma"

_jpm, 2023, doi:10.3390/jpm13050767_

Round 1

Reviewer 1 Report

This study seeks to establish a potential prognostic value for hypoxia markers and HPV status in SNSCC. Although in the literature HPV infection has been associated with better, and hypoxia with worse overall survival, the authors did not find this in their analysis on 40 cases SNSCC. Only hypoxia marker CA-IX appeared correlated with poor overall survival.

Although the study brings new and original data to the study of SNSCC, the presentation needs improvement, with better justification of the rationale and of the methodology. Please find below some concrete remarks.

Major comment

The Introduction begins to state that hypoxia is promising as a therapeutic target and may play a role in radio/chemo-resistance. So the reader expects that this study will investigate hypoxia markers in relation to response to radio/chemotherapy, but the Introduction continues with HPV as a factor to be studied in relation to the hypoxia markers. This is not well explained or justified, the subject of HPV in SNSCC needs more introduction. Only at the end of Discussion the authors discuss the possible link between hypoxia and HPV, which consists in their opposite association with prognosis and with radioresistance.

Would it be possible for the authors to re-analyze their data and look for correlation of hypoxia markers and HPV with survival (both progression-free and overall), taking into account the type of treatment as detailed in Table 2? Such a multivariate analysis could provide info on radio/chemo-resistance and make the study more interesting.

Minor comments

1. The Abstract states that SNSCC has recently been associated with hypoxia-induced cellular changes. But I could find no such statement in the body of the text. Maybe it was meant HNSCC? And what is meant by 'cellular changes'?

2. Introduction: '...incidence of 35-88%'. What is meant here? SNSCC represent 35-88% of all sinonasal cancers?

3. Table 1 has 40 patients but Table 2 has 39...?

4. M+M states a minumum follow-up of 18 months, it would be better to give the mean and median follow-up time.

5. M+M should explain that the HPV analysis is DNA based (it is stated in the Discussion..) and refer more completely to the supplier.

6. Can SCC of vestibule be considered SNSCC or is it a skin SCC?

7. With 67.5% of patients having a T4 stage tumor, and with >18 months (I see in figure 2 up to 75 months) follow-up, did really only 3 of 39 patients develop a recurrence? With how much progression-free time? And did only 15/39 die of disease? Both figures seem low and not in accord with the statement that 'SNSCC is an aggressive malignancy'... Or could it be explained by the fact that follow-up time of many patients was very short?

8. Why did the authors not also analyze the association between HPV and progression-free and overall survival?

9. Discussion: Paragraph 2 first sentence and paragraph 3 second sentence state the same info.

Author Response

The Introduction begins to state that hypoxia is promising as a therapeutic target and may play a role in radio/chemo-resistance. So the reader expects that this study will investigate hypoxia markers in relation to response to radio/chemotherapy, but the Introduction continues with HPV as a factor to be studied in relation to the hypoxia markers. This is not well explained or justified, the subject of HPV in SNSCC needs more introduction. Only at the end of Discussion the authors discuss the possible link between hypoxia and HPV, which consists in their opposite association with prognosis and with radioresistance.

As suggested by the referee, the “Introduction” section has been modified as follows:

Line 52

“ In this context, it seems important to investigate potential biological mechanisms that may be associated with resistance to chemo/radiotherapy. Tumor hypoxia has been shown to have a negative impact on chemo/radiosensitivity and on survival in several tumor types, including HNSCC. [5] In fact, the lack of oxygen induces a reduction of reactive oxygen species, significantly decreasing the DNA damage caused by radiotherapy, and also increases proliferation of anarchist blood microvessels causing irregular drug diffusion. [6] Although significant progress has been made in the understanding of the biology of sinonasal cancers [7], very few studies have been published on the hypoxic changes in SNSCC. [8]

Several factors are involved in the cellular response to hypoxia, and can be assessed using standard immunohistochemistry procedures. ”

Line 72

 “On the other hand, Human papillomavirus (HPV) positive status is associated with a better response to radiotherapy and with better survival outcomes in HNSCC, especially in the oropharynx. [11,12] The implication of HPV in the pathogenesis of SNSCC has already been reported. [13] A recent study based on the Surveillance, Epidemiology, and End Results Program (SEER) database even found an increase in the incidence of HPV-associated SNSCC and prevalence of HPV-positive SNSCC over recent decades. [14] The prognostic significance of HPV-status in SNSCC has remained uncertain for a long time, but it now seems that HPV positivity is associated to better overall survival. [11, 15] The potential association between hypoxia and HPV status has however never been explored.”

Would it be possible for the authors to re-analyze their data and look for correlation of hypoxia markers and HPV with survival (both progression-free and overall), taking into account the type of treatment as detailed in Table 2? Such a multivariate analysis could provide info on radio/chemo-resistance and make the study more interesting.

We thank both referees for this important comment. Indeed, a multivariate analysis would have made it possible to investigate the specific impact of tumor hypoxia on the response to chemotherapy and radiotherapy. Unfortunately, the size of the cohort (40 patients) and the heterogeneity of the treatment strategies (table 2) precluded any multivariate analysis. The following sentence has been added in the “limitation” section of the discussion to address this issue:

Line 267 “This study has several limitations: first, the retrospective nature of the data; second, the relatively limited size of the cohort and the heterogeneity of treatment strategies, that did not allow a multivariate analysis”.

Minor comments

  1. The Abstract states that SNSCC has recently been associated with hypoxia-induced cellular changes. But I could find no such statement in the body of the text. Maybe it was meant HNSCC? And what is meant by 'cellular changes'?

We thank the referee for this comment. To clarify this point, the Abstract was modified as follows:

“Background: In head and neck squamous cell carcinoma, tumor hypoxia has been associated with radio/chemoresistance and poor prognosis, whereas Human papillomavirus (HPV)-positive status has a positive impact on treatment response and survival outcomes. The aim of this study was to evaluate the expression and the potential prognostic value of hypoxia-induced endogenous markers in patients treated for squamous cell carcinoma of the nasal cavity and paranasal sinuses (SNSCC) and their correlation with HPV status.”

  1. Introduction: '...incidence of 35-88%'. What is meant here? SNSCC represent 35-88% of all sinonasal cancers?

Answer: Indeed, 35-58% was not the incidence rate. The sentence has been modified as follows:

“Sinonasal squamous cell carcinoma (SNSCC) is the most common sinonasal malignancy, making up approximately 60% of cases; [1] an incidence rate of 0.36 cases per 100,000 population per year has been reported. [2]”

  1. Table 1 has 40 patients but Table 2 has 39...?

The error has been corrected in Table 2.

  1. M+M states a minumum follow-up of 18 months, it would be better to give the mean and median follow-up time.

The following sentence has been added:

Line 134 “The mean and median follow-up were 28.4 months (SD 19.8, 18-77 months) and 27, respectively”

  1. M+M should explain that the HPV analysis is DNA based (it is stated in the Discussion..) and refer more completely to the supplier.

The Material and methods section has been modified as follows:

“HPV status

HPV genotyping was used to assess HPV status. Formalin-fixed, paraffin-embedded (FFPE) biopsies were deparaffinized, and HPV detection and genotyping was carried out by multiplex real-time PCR using the Anyplex II HPV28 Detection assay (Seegene, Seoul, Republic of Korea). This assay allows a simultaneous detection and identification of 28 HPV types including 13 high-risk.”

  1. Can SCC of vestibule be considered SNSCC or is it a skin SCC?

Considering the distinct behavior of vestibule SCC, closer to that of a skin SCC, we made the choice to exclude this localization from the analysis.

  1. With 67.5% of patients having a T4 stage tumor, and with >18 months (I see in figure 2 up to 75 months) follow-up, did really only 3 of 39 patients develop a recurrence? With how much progression-free time? And did only 15/39 die of disease? Both figures seem low and not in accord with the statement that 'SNSCC is an aggressive malignancy'... Or could it be explained by the fact that follow-up time of many patients was very short?

Indeed, there were not only 3 recurrences: altogether at last follow-up, 17/40 patients (42.5%) had died of disease and 3/40 were alive with disease (7.5%). The relatively short follow-up might explain these low figures. Table 2 has been modified to clarify this, and the following sentence has been added in the Discussion section:

“This study has several limitations: first, the retrospective nature of the data; second, the relatively limited size of the cohort and the heterogeneity of treatment strategies, that did not allow a multivariate analysis; third, the short follow-up (28.4 months), that may have an impact on the survival analysis.”

  1. Why did the authors not also analyze the association between HPV and progression-free and overall survival?

The study was mainly focused on the assessment of hypoxic markers, but it is indeed also possible to analyze the association between HPV and survival rates: the following sentence has been added in the results section, as well as a supplementary figure.  

“There was no statistical correlation between HPV status and OS / LRRFS (p= 0.62 and 0.43, respectively; Supplementary figure 4).”

  1. Discussion: Paragraph 2 first sentence and paragraph 3 second sentence state the same info.

This error has been corrected.

Reviewer 2 Report

The purpose of this study was to investigate Expression, prognostic value and correlation with HPV status of hypoxia-induced markers in sinonasal squamous cell carcinomaA total of 40 patients treated in a curative intent for Sinonasal squamous cell carcinoma (SNSCC) were included in the study tween January 2009 and December 2020. Small number of cases and long time factor may lead to large bias. Other details need further improvement. 

1. The TNM staging according to the 8th UICC was not mentioned in the article or references, but according to the 8th AJCC,N stage was with N0-3,b the N stage showed in table1 without N2 and N3, is it true? Was all patient Metastasised in a single ipsilateral lymph node, and the greatest dimension was smaller than 3cm and without ENE, the data was collected from 2009 to 2020, over 10 years, the result was so strange, please comfirm the clinical data.

2. Induction CT was mentioned in the table2 did, but not in treatment modality part, can you show us what kind of patients should carried out with IC+RT, and IC+S+Adjuvant RT?

3. The result showed in the article, Tumors with high CA-IX expression correlated with poor OS, but other markers (like GLUT-1, HIF-1α, VEGF, VEGF-R1, also were detected at high level) with no correlation with OS, can you show us all the figures.

4. VEGF has been related to poor prognosis, nodal metastasis, and low survival in head and neck squamous cell carcinoma, But the observations from this study was with opposite viewpoint, please explain why?

5. Whether CA IX is an independent prognostic factor for OS, it will affects the reliability of the results.

6. The correlation between HPV and the marker of hypoxia pathway is the main conclusion, and it is recommended to add more graphs to improve the persuasive effect.

Author Response

Reviewer 2

The purpose of this study was to investigate Expression, prognostic value and correlation with HPV status of hypoxia-induced markers in sinonasal squamous cell carcinoma. A total of 40 patients treated in a curative intent for Sinonasal squamous cell carcinoma (SNSCC) were included in the study between January 2009 and December 2020. Small number of cases and long time factor may lead to large bias. Other details need further improvement. 

  1. The TNM staging according to the 8th UICC was not mentioned in the article or references, but according to the 8th AJCC,N stage was with N0-3,b the N stage showed in table1 without N2 and N3, is it true? Was all patient Metastasised in a single ipsilateral lymph node, and the greatest dimension was smaller than 3cm and without ENE, the data was collected from 2009 to 2020, over 10 years, the result was so strange, please comfirm the clinical data.

The original data were rechecked: indeed, there had been a confusion when retrieving the data from the original database (0=N0 and 1=N+). This error has been corrected in the “Results” section in Table 1. Still, the N+ rate was 7.5% (1 N1, 1 N2a, 1 N2B), which is a little less than other series. The fact that TNM staging was performed according to the 8th UICC has also been added in the “Materials and Methods” section.

  1. Induction CT was mentioned in the table2 did, but not in treatment modality part, can you show us what kind of patients should carried out with IC+RT, and IC+S+Adjuvant RT?

We thank the referee for this important comment. To clarify this, the following sentence has been added in the “Materials and Methods” section:

“Neoadjuvant chemotherapy was considered in locally advanced tumors that were not amenable to surgery or radiotherapy at the outset, with the aim of achieving tumor deflation and facilitating surgery or chemoradiotherapy. In case of partial/complete response to neoadjuvant chemotherapy, chemoradiotherapy was considered; in case of stable or progressive disease, surgery was considered, or if this was impossible, chemoradiotherapy.

  1. The result showed in the article, Tumors with high CA-IX expression correlated with poor OS, but other markers (like GLUT-1, HIF-1α, VEGF, VEGF-R1, also were detected at high level) with no correlation with OS, can you show us all the figures.
  2. The correlation between HPV and the marker of hypoxia pathway is the main conclusion, and it is recommended to add more graphs to improve the persuasive effect.

As required, the other graphs have been added (as Supplementary figures when the results were not statistically significant).

  1. VEGF has been related to poor prognosis, nodal metastasis, and low survival in head and neck squamous cell carcinoma, But the observations from this study was with opposite viewpoint, please explain why?

As pointed by the referee, the results of our study are in contrast with other findings in HNSCC. The survival curves displayed in Supplementary Figure 3 even suggests that VEGF overexpression was associated with a trend towards better OS (although not significant). To underline this issue, the following sentences have been added in the “Discussion” section:

“VEGF (…) has been related to poor prognosis, nodal metastasis, and low survival in head and neck squamous cell carcinoma. [25, 30] This is in contrast with the results of the current study. This may be explained by the fact that the relatively small size of the cohort limits the power of the study. Of note, although a recent meta-analysis by Zang et al. [29] found that VEGF overexpression was a poor predictor for overall and disease-free survival in head and neck squamous cell carcinoma, the authors underlined that these results were obtained only using univariate analysis. Previous studies have also failed to confirm the association between VEGF expression and survival outcomes. [31]”

  1. Whether CA IX is an independent prognostic factor for OS, it will affects the reliability of the results.

We thank both referees for this important comment. Indeed, a multivariate analysis would have made it possible to investigate the specific impact of tumor hypoxia on the response to chemotherapy and radiotherapy. Unfortunately, the size of the cohort (40 patients) and the heterogeneity of the treatment strategies (table 2) precluded any multivariate analysis. The following sentence has been added in the “limitation” section of the discussion to address this issue:

Line 262 “This study has several limitations: first, the retrospective nature of the data; second, the relatively limited size of the cohort and the heterogeneity of treatment strategies, that did not allow a multivariate analysis”.

Round 2

Reviewer 1 Report

The authors have done justice to many of the comments and the paper has gained more clarity. However, one points needs to be addressed.

Major comment

Upon comment 6, the authors answered: "Considering the distinct behavior of vestibule SCC, closer to that of a skin SCC, we made the choice to exclude this localization from the analysis." From this I understand that 6 cases would be excluded from this study, leaving 34 cases SNSCC. However, all presented data still include all 40 cases and all the analyses and all results remain the same....?

Minor comment

Abstract "Conclusion: This study provides 36 data on the expression of hypoxia-induced endogenous markers in patients treated for SNSCC and 37 underlines the potential role of CA-IX as biomarker of tumor aggressiveness in SNSCC." I believe more appropriate would be to state "... underlines the potential role of CA-IX as a prognostic biomarker in SNSCC."

Author Response

Reviewer 1

Upon comment 6, the authors answered: "Considering the distinct behavior of vestibule SCC, closer to that of a skin SCC, we made the choice to exclude this localization from the analysis." From this I understand that 6 cases would be excluded from this study, leaving 34 cases SNSCC. However, all presented data still include all 40 cases and all the analyses and all results remain the same....?

Answer: We thank the referee for this comment. When the data were retrieved from the clinical files, detailed information on tumor localization was collected when available. This information was displayed in table 1, with “vestibule” for tumor extending to the nasal vestibule, or “septum” for tumors invading specifically the septum, and “nasal fossa” when there was no specific information on tumor extensions. Actually all of these were nasal fossa tumors and we confirm that no cases of SCC arising primarily from the skin of the nasal vestibule was included in the study. To clarify this, table 1 has been modified as follows, pooling “septum” and “vestibule” with “nasal fossa”:

Minor comment

Abstract "Conclusion: This study provides data on the expression of hypoxia-induced endogenous markers in patients treated for SNSCC and underlines the potential role of CA-IX as biomarker of tumor aggressiveness in SNSCC." I believe more appropriate would be to state "... underlines the potential role of CA-IX as a prognostic biomarker in SNSCC."

Answer: Following your comment we have modified the manuscript accordingly (see abstract section).

Reviewer 2 Report

1. In response to comment 5, “Of note, although a recent meta-analysis by Zang et al. [29] found that VEGF overexpression was a poor predictor for overall and disease-free survival in ……….”. “a poor predictor” means the predicting potency of VEGF is poor, while Zang et al. suggested that VEGF overexpression was significantly associated with worse OS and DFS. Additionally, this study was published 10 years ago, not recently.

2. The authors did not address my concern 4 “The correlation between HPV and the marker of hypoxia pathway is the main conclusion, and it is recommended to add more graphs to improve the persuasive effect.” Supplementary figures 1-4 have been added as recommended in my comment 3, but these are not responses for comment 4.

3. The authors’ responses are unable to address my concern 6 sufficiently.

Author Response

Reviewer 2

  1. In response to comment 5, “Of note, although a recent meta-analysis by Zang et al. [29] found that VEGF overexpression was a poor predictor for overall and disease-free survival in ……….”. “a poor predictor” means the predicting potency of VEGF is poor, while Zang et al. suggested that VEGF overexpression was significantly associated with worse OS and DFS. Additionally, this study was published 10 years ago, not recently.

Answer: Accordingly, this sentence has been modified as follows:

Line 247-250 “Of note, although a meta-analysis by Zang et al. [29] found that VEGF overexpression was significantly associated with worse OS and DFS in head and neck squamous cell carcinoma (…)”

  1. The authors did not address my concern 4 “The correlation between HPV and the marker of hypoxia pathway is the main conclusion, and it is recommended to add more graphs to improve the persuasive effect.” Supplementary figures 1-4 have been added as recommended in my comment 3, but these are not responses for comment 4.

Answer: As stated in the Materials and methods section, “The Chi-square test or Fisher exact test for independent proportions was performed to evaluate differences and similarities in expression of CA-IX, GLUT-1, VEGF, VEGF-R1, HIF-1α, and in HPV status”. As a consequence, there was no graphical representation of the association between HPV and markers of hypoxia, and we only stated that no statistical association had been found (p>0,05).

Following your comment, the information regarding the correlation between HPV and hypoxic markers has been modified in line 256-261:

“The link between HPV status and hypoxia remains however unclear. [12, 14] Clinical data suggest that there is no significant difference in the level, nor distribution of hypoxia in HPV-positive and HPV-negative head and neck squamous cell carcinoma, as measured by a 15-gene hypoxia classifier and 18F-MISO PET, [33,34] This is in line with our results, which suggest that there is no correlation between HPV status and hypoxic markers in SNSCC”

  1. The authors’ responses are unable to address my concern 6 (“Whether CA IX is an independent prognostic factor for OS, it will affects the reliability of the results”) sufficiently.

Answer: As stated in our previous response, the size of the cohort (40 patients) and the heterogeneity of the treatment strategies (table 2) unfortunately precluded any multivariate analysis. We completely agree that results obtained using univariate analysis must be considered cautiously. However, we also think that the data provided in this paper may add information to the current knowledge on SNSCC. To be clearer about the limits of the statistical analyses, the following sentences have been added in the abstract, discussion and conclusion:

Line 33-35 “High CA-IX expression was associated in univariate analysis with poor OS (p=0.035), but there was no significant association between GLUT-1, VEGF, VEGF-R1 and HIF-1α expression and OS/LRRFS”

Line 200 “CA-IX overexpression was associated with poor OS in univariate analysis”

Line 262 “This study has several limitations: first, the retrospective nature of the data; second, the relatively limited size of the cohort and the heterogeneity of treatment strategies, that did not allow a multivariate analysis”.

Line 290 “High CA-IX expression was associated with poor OS (p=0.035) in univariate analysis”

Round 3

Reviewer 1 Report

I have no more comments, thank you.